

# The relationship between egg size and helper number in cooperative breeders: a meta-analysis across species

Tanmay Dixit[1], Sinead English[1,2] and Dieter Lukas[1,3]

[1] Department of Zoology, University of Cambridge, Cambridge, United Kingdom
[2] Current affiliation: School of Biological Sciences, University of Bristol, Bristol, United Kingdom
[3] Current affiliation: Department of Human Behavior, Ecology and Culture, Max Plank Institute for Evolutionary Anthropology, Leipzig, Germany

## ABSTRACT

**Background**. Life history theory predicts that mothers should adjust reproductive investment depending on benefits of current reproduction and costs of reduced future reproductive success. These costs and benefits may in turn depend on the breeding female's social environment. Cooperative breeders provide an ideal system to test whether changes in maternal investment are associated with the social conditions mothers experience. As alloparental helpers assist in offspring care, larger groups might reduce reproductive costs for mothers or alternatively indicate attractive conditions for reproduction. Thus, mothers may show reduced (load-lightening) or increased (differential allocation) reproductive investment in relation to group size. A growing number of studies have investigated how cooperatively breeding mothers adjust pre-natal investment depending on group size. Our aim was to survey these studies to assess, first, whether mothers consistently reduce or increase pre-natal investment when in larger groups and, second, whether these changes relate to variation in post-natal investment.

**Methods**. We extracted data on the relationship between helper number and maternal pre-natal investment (egg size) from 12 studies on 10 species of cooperatively breeding vertebrates. We performed meta-analyses to calculate the overall estimated relationship between egg size and helper number, and to quantify variation among species. We also tested whether these relationships are stronger in species in which the addition of helpers is associated with significant changes in maternal and helper post-natal investment.

**Results**. Across studies, there is a significant negative relationship between helper number and egg size, suggesting that in most instances mothers show reduced reproductive investment in larger groups, in particular in species in which mothers also show a significant reduction in post-natal investment. However, even in this limited sample, substantial variation exists in the relationship between helper number and egg size, and the overall effect appears to be driven by a few well-studied species.

**Discussion**. Our results, albeit based on a small sample of studies and species, indicate that cooperatively breeding females tend to produce smaller eggs in larger groups. These findings on prenatal investment accord with previous studies showing similar load-lightening reductions in postnatal parental effort (leading to concealed helper effects), but do not provide empirical support for differential allocation. However, the considerable variation in effect size across studies suggests that maternal investment is mitigated by additional factors. Our findings indicate that variation in the social

Corresponding author
Tanmay Dixit, tanmay.dixit@cantab.net

environment may influence life-history strategies and suggest that future studies investigating within-individual changes in maternal investment in cooperative breeders offer a fruitful avenue to study the role of adaptive plasticity.

## INTRODUCTION

Life history theory suggests that, to maximize their lifetime reproductive success, breeders should adjust their reproductive investment according to both current and future conditions (*Stearns, 1992*). There is considerable evidence that increased maternal investment into offspring can strongly and positively affect offspring development, survival and future chances of reproduction (*Bolton, 1991*; *Lindström, 1999*; *Krist, 2011*), could be but can also negatively influence maternal survival and investment into future reproductive attempts (*Lack, 1947*; *Maynard Smith, 1977*; *Berube, Festa-Bianchet & Jorgenson, 1996*; *Metcalfe & Monaghan, 2001*; *Lummaa & Clutton-Brock, 2002*; *Hanssen et al., 2005*). A mounting body of studies has shown that mothers adaptively alter investment into reproduction via offspring traits, such as egg size or hormone provision into eggs, depending on breeding conditions, such as food availability or mate quality and contribution to care (e.g., *Cunningham & Russell, 2000*; *Eising et al., 2001*; *Gil et al., 2004*; *Bolund, Schielzeth & Forstmeier, 2009*). In addition, in group-living species, mothers might adjust investment according to other salient features of their social environment besides their partner, in particular if there is variation in the presence or expected helping contribution of alloparental carers.

Cooperatively breeding species provide an opportunity to study the changes in parental investment in response to variability in the social group environment. In these species, helpers assist breeding individuals with caring for offspring (*Stacey & Koenig, 1990*; *Clutton-Brock, 2002*; *Koenig & Dickinson, 2004*). Differences in the number of additional group members, both within and among species, generate natural variation in the social environment mothers experience (*Russell & Lummaa, 2009*). Given the trade-offs between investment into the current brood and later reproductive attempts, investments by mothers into the current brood are expected to vary in the presence of helpers (*Crick, 1992*; *Hatchwell, 1999*; *Savage, Russell & Johnstone, 2013*). Detailed field observations of various cooperative breeders, particularly in birds, have presented evidence that the amount of maternal investment at the nest (provisioning) is associated with variation in the workforce present when breeding (*Hatchwell & Russell, 1996*; *Woxvold, Mulder & Magrath, 2006*; *Cockburn et al., 2008*; *Russell et al., 2008*). Associations between maternal investment and helper numbers might result from differences in territory quality or maternal condition (*Cockburn et al., 2008*; *Russell & Lummaa, 2009*). In at least some instances, however, reduced maternal provisioning may be interpreted as an adaptive response of females to variation in the social group environment, such that breeding females maximize their
lifetime fitness by reducing post-natal investment in current offspring when helpers compensate for such a reduction (*Wright, 1998*; *Russell et al., 2008*; *McDonald, Kazem & Wright, 2009*).

Social influences could also influence maternal pre-natal investment (*Savage, Russell & Johnstone, 2015*): in cooperative breeders, altered maternal investment into eggs might be compensated for or enhanced by the protection, provisioning or incubation of offspring provided by helpers subsequent to laying (*Russell et al., 2007*; *Taborsky, Skubic & Bruintjes, 2007*; *Russell & Lummaa, 2009*). One hypothesis that has been proposed, the primary one we test here, is that cooperatively breeding mothers will show decreased investment into egg size when helpers are present to compensate for such reduced investment: the 'load-lightening hypothesis' (here onwards, LL hypothesis) (*Russell & Lummaa, 2009*; *Savage, Russell & Johnstone, 2015*; also sometimes referred to as 'concealed helper effects', *Koenig, Walters & Haydock, 2009*). This hypothesis argues that, since egg production is costly in terms of future maternal survival and reproductive success (*Visser & Lessells, 2001*; *Williams, 2005*), lifetime reproductive success will be increased if reductions in egg size have little or no effect on offspring fitness owing to the compensation provided by helpers (*Savage, Russell & Johnstone, 2013*). Observations that egg sizes decrease with an increased number of helpers at the nest would provide support for the LL hypothesis (*Taborsky, Skubic & Bruintjes, 2007*; *Russell & Lummaa, 2009*), though such an association could also reflect factors that influence pre- and post-natal conditions independently or changes in helper behaviour in response to maternal investment (*Savage, Russell & Johnstone, 2015*).

An alternative hypothesis is that mothers increase investment in larger groups in order to take advantage of good conditions: the so-called 'differential allocation (DA) hypothesis' (*Sheldon, 2000*). Usually applied to cases when females have a particularly attractive mate (*Sheldon, 2000*; *Horváthová, Nakagawa & Uller, 2011*), it has been expanded to cooperative breeding, predicting that if a female is assisted by a large numbers of helpers (indicating good breeding conditions), she should increase investment due to higher current potential reproductive success or the increased reproductive value of the current brood (*Valencia et al., 2006*; *Russell & Lummaa, 2009*; *Savage, Russell & Johnstone, 2015*). Thus, the DA hypothesis, unlike the LL hypothesis, would predict an increase in egg investment with increased number of helpers.

Variation in prenatal maternal investment is likely to be associated with the care offspring receive post-natally (*Savage, Russell & Johnstone, 2015*). Here we focus on two factors that might shape the relationship between prenatal maternal investment and postnatal offspring care: mothers are expected to reduce investment in egg size if such a reduction can be directly substituted by helper care (helpers provide food, and helper efforts can buffer a small size early in life) and if there are no benefits to offspring of receiving investment beyond a certain threshold (for example if offspring need to reach a certain body size to increase their survival, but additional increments in size do not lead to higher offspring fitness). The interactions between mothers and helpers during the post-natal stage might provide information on these two factors. First, if prenatal investment can be substituted by postnatal care, we would predict that mothers show consistent reductions in their pre-

and postnatal investment whereas if prenatal investment primes offspring to benefit from increased postnatal help, maternal postnatal care behaviour should be independent of the presence of helpers. Accordingly, we predict that mothers are more likely to produce smaller eggs in groups with helpers in species in which mothers also reduce their postnatal investment in the presence of helpers. Second, if additional investments continue to provide benefits, helping efforts are expected to increase linearly with group size, whereas if there is a limit up to which investment is beneficial, helping efforts might plateau. Therefore, we predict that mothers are more likely to produce smaller eggs in larger groups in species where total helping efforts plateau and do not increase linearly with group size.

In this study, we perform a meta-analysis to determine whether the current evidence indicates that variation in maternal prenatal investment in the form of egg size systematically varies with the number of helpers across cooperative breeders, and whether the direction of the relationship indicates whether load-lightening or differential allocation might be more prevalent. Our study systematically combines the recently available observations to assess whether consistent increases or decreases in maternal investment at the egg stage in response to helper numbers have been observed and test predictions that changes in maternal pre-natal investment are associated with the post-natal care offspring experience.

## MATERIALS & METHODS

A literature search was conducted to find relevant papers in which egg investment traits and helper number were measured. We focused on egg size as our measure of maternal investment as it is clearly under maternal control while subsequent investment into offspring is frequently shared among mothers, fathers, and helpers. As such, we decided not to include data on mammals since offspring size at birth is not available for most cooperatively breeding mammals and lactation makes it more difficult to ascertain when investment is fully under maternal influence versus that of helpers. For the purpose of this analysis, cooperative breeders were defined as species where additional group members act as allomaternal helpers and provide resources that can replace and hence compensate for adjustments in maternal investment, such that mothers could potentially alter their investment into egg size based on helper number. Studies providing a quantitative measure of the link between maternal investment and group size in eusocial species are rare. Moreover, we did not specifically search for these as our focus was on instances where maternal investment could vary both pre- and post-natally: in eusocial species, mothers tend not to provide any offspring care beyond egg production when helpers are present and almost all post-natal care is performed by helpers (*Keller & Chapuisat, 2010*). We also did not include species where offspring care is shared only among reproductively active females (e.g., *Grinsted, Breuker & Bilde, 2014*), as it is difficult to determine how individual mothers differ in their investment. On the other hand, cooperatively polyandrous species were included: while additional group members (in this case, males) may share paternity, efforts by the males represent alloparental care and could replace female investment into the provisioning and protection of offspring (*Davies, 1985*).

Six relevant papers (*Legge, 2000*; *Taborsky, Skubic & Bruintjes, 2007*; *Koenig, Walters & Haydock, 2009*; *Canestrari, Marcos & Baglione, 2011*; *Santos & Macedo, 2011*; *Lejeune et al.,*

*2016*) were located using keyword searches on Google Scholar with the search terms 'egg size' and 'helpers', on 30th October 2016 (search carried out by TD and confirmed by DL; search criteria were specific and hence disagreements were solved easily by discussion). Forward and backward citation searches were then performed to find five further relevant studies (*Russell et al., 2007*; *Paquet et al., 2013*; *Santos & Nakagawa, 2013*; *Santos, 2016*; *Valencia et al., 2016*). One study (*Langmore et al., 2016*) was identified as an 'online-early' abstract during the literature search, and was included as full publication occurred during the course of our study. Relevant studies were defined as those that investigated a relationship between egg size—measured specifically as egg mass or egg volume—and number of helpers (determined by reading abstracts and figures). Studies were then included if they provided the required test-statistic for use in meta-analysis (see below), or authors were contacted for raw data if suitable data were not provided in the paper. Where a direct measure of egg volume was not available, this was calculated from length and width using Hoyt's formula (*Hoyt, 1979*).

In seven studies, the change in maternal investment per helper was not provided, rather comparisons were given between investment when breeding in groups versus pairs. In these cases, and in cases where data on egg size traits with helpers were referred to but not provided, authors were contacted to provide the raw data for number of helpers in each group, and linear models, using the command lme in the package 'nlme' (*Pinheiro et al., 2015*) in RStudio version 0.99486 (*RStudio Team, 2015*) using R version 3.2.2 (*R Core Team, 2015*), were used to calculate test statistics. The test statistics—$F$, $t$ and $\chi^2$—were converted to the statistic '$r$' using formulae provided by *Rosenthal (1994)*, *Lajeunesse (2013)* (in *Koricheva, Gurevitch & Mengersen, 2013*), and *Rosenberg (2010)*. Standard formulae were then $Z$-transformed, controlling for the asymptotic behaviour of the statistic $r$. We followed the detailed advice in *Jennions, Moller & Petrie (2001)* for these methods, and formulae therein for calculations of effect sizes, transformations, and variances.

We also investigated whether any relationship between maternal pre-natal investment and group size varied according to how aspects of post-natal care depend on helper number (specifically, whether mothers reduce post-natal care in the presence of helpers, and whether helpers provide benefits to offspring additively or up to a threshold). We thus searched for additional data on each species on the relationship between maternal post-natal care, helper effects on offspring and group size. Specifically, for each data point, we recorded whether the amount of support mothers provide to offspring (in most studies measured as the number of feeds per hour) is significantly lower when at least one helper (of that sex, in instances where the original study provided separate information for each sex on the relationship between egg size and helper number) is present than when groups only consist of mothers and their mate. To determine whether helping effort increases linearly or plateaus with group size (all studies we included showed either of these two options), we checked which function authors had used to model the relationship between group size and total amount of care provided at the nest (measured as number of feeds per hour in bird species, and as egg caring in *Neolamprologus pulcher*), inspected the associated graphs to see whether increases of a second, third, etc. helper was similar to the increase observed in groups with just a single helper, and read through the associated paper to check

if the description by the authors confirmed the graphical pattern (for example if they stated that the amount of support increased linearly with group size).

Functions in the R package 'metafor' (*Viechtbauer, 2010*) were used to compute heterogeneity and average effect sizes, and to create funnel plots to visualise the data. Cochran's Q-test was used to test heterogeneity between effect sizes in all analyses carried out. Heterogeneity describes the variation in the data which can be attributed to actual variation between species, variation in study design, or publication bias (*Nakagawa & Santos, 2012*). The average effect size describes the overall effect of helpers on changes in maternal investment into eggs, and hence indicates the strength and direction of the effect, determining whether the LL or DA hypothesis receives more empirical support. Contour-enhanced funnel plots were used to visualise effect sizes, and precision was defined as 1/SE where $SE = \sqrt{\text{(variance)}}$. In this study, with high heterogeneity predicted, a contour plot would suggest a lack of publication bias if the outliers occurred at all precision values, and not solely at low precision (*Macaskill, Walter & Irwig, 2001*). Publication bias was also quantified using Egger's regression test (*Egger et al., 1997*) on the dataset, using the regtest function in the metafor package, and sample standard error as a predictor (*Viechtbauer, 2010*). We followed suggested best practices to minimize bias in the dataset: the literature search was comprehensive, and authors were contacted if published statistics were insufficient to compute effect sizes (*Gates, 2002*). However, our sample remains small, with high heterogeneity since authors might have tested hypotheses with opposite effects (negative for LL and positive for DA), such that we are aware that any conclusions are likely to remain limited at this stage.

A multilevel linear model with restricted maximum likelihood estimation (REML) analysis was carried out on all effect sizes, controlling for study as a random factor, to take into account the non-independence of individual effect sizes calculated from the same study. In several of the studies, authors reported multiple relationships: running separate analyses on male and female helpers, reporting data on both egg size and mass, using different statistical models, considering variation among or within individual females, or considering different mating systems. Each study focussed on a single species, and in all but two cases, studies considered different species. The exceptions were *Paquet et al. (2013)* and *Santos (2016)*, both of which studied sociable weavers; and *Russell et al. (2007)* and *Langmore et al. (2016)*, both of which studied superb fairy-wrens. As all of the studies in our sample had slightly different methodologies, we initially controlled for study rather than species as a random term as we deemed this a more parsimonious measure for data non-independence. We also conducted a separate analysis controlling for species. Given the small sample size (10 species), we decided not to include phylogenetic relatedness among species as a potential covariate.

To investigate the association between the post-natal behaviour of mothers and of helpers with the relationship between pre-natal maternal investment and group size, additional analyses were carried out on all effect sizes, using the type of behaviour ('mothers decrease postnatal investment (in groups with on helper compared to pairs)' versus 'mothers do not decrease postnatal investment (no change or increase when a helper is present)'; 'helper benefits increase linearly (with group size)' versus 'helper benefits plateau') as a moderator

(i.e., predictive variable). Separate multilevel linear mixed effects models, with REML analyses, were conducted for each post-natal response trait (maternal response and helping effort pattern), controlling for species or for study. Following these analyses, in order to examine the difference among species more intuitively, we carried out separate multilevel linear models with REML analysis, controlling for species, on the subset of the data where mothers reduce postnatal care, on the subset of the data where mothers do not reduce postnatal care, on the subset of the data where helping efforts increase linearly with group size, and on the subset of the data where helping efforts plateau with group size.

## RESULTS

Twenty effect sizes from 12 studies of 10 species (nine bird species; one fish species) were extracted from publications or calculated from raw data (see Table S1). There was no evidence of publication bias using Egger's regression test (test value $Z = -1.03$; $p = 0.30$). In addition, the funnel plot was approximately symmetrical (Fig. 1), particularly for low values of 1/SE (i.e., low precision), suggesting little publication bias. As visible on the funnel plot (Fig. 1), the overall effect size (when controlling for study) was negative and significantly different from zero (effect size $= -0.180$, $p = 0.03$, upper and lower confidence limits $= -0.023$ and $-0.338$; $n = 20$ effect sizes from 12 studies). The Cochran's Q-test value was highly significant ($Q = 128$ on 19 degrees of freedom, $p < 0.001$), suggesting that the data are highly heterogeneous. When the same dataset was analysed controlling for species rather than study, the effect size was negative and marginally not significant (effect size $= -0.164$, $p = 0.08$, upper and lower confidence limits $= 0.020$ and $-0.348$; $n = 20$ effect sizes from 10 species).

When we examined how the relationship between pre-natal investment (egg size) and helper number varied depending on how mothers adjust post-natal investment in larger groups, we found that—in analyses controlling for study—egg sizes are significantly more likely to be reduced in the presence of more helpers in species in which mothers also reduce their post-natal investment in the presence of helpers (estimate ['mothers decreases postnatal investment in larger groups' versus 'mothers do not decrease postnatal investment'] $= -0.334 \pm 0.158$, $p = 0.03$, upper and lower confidence limits $= -0.025$ and $-0.644$; $n = 20$ effect sizes from 12 studies). A similar, but non-significant, relationship between post-natal maternal response and the relationship between egg size and helper number is found when controlling for species rather than study (estimate [reduction in postnatal investment present versus not present] $= -0.336 \pm 0.178$, $p = 0.06$, upper and lower confidence limits $= 0.012$ and $-0.685$ $n = 20$ effect sizes from 10 species). The overall effect size (i.e., of the relationship between helper number and egg size) for those studies where mothers reduce their post-natal investment in the presence of helpers was significantly less than zero controlling for species identity (effect size $= -0.274$, $p = 0.02$, upper and lower confidence limits $= -0.046$ and $-0.503$, $n = 13$ effect sizes from 7 species) (Fig. 2A), with significant heterogeneity among values ($Q = 98.2$ on 12 degrees of freedom, $p < 0.001$). There are only very few investigations of effects where mothers do not reduce their post-natal investment in the presence of helpers, restricted to three species, but the

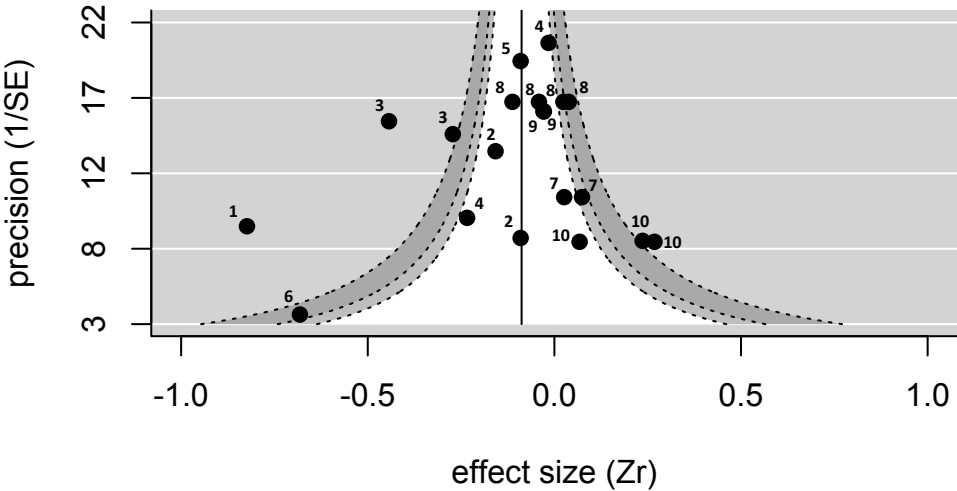

**Figure 1** **A funnel plot showing the 20 effect sizes extracted from 12 studies on 10 species.** The pooled estimate (solid line), controlling for study, is significantly smaller than zero. The funnel shows the regions where 90% (white), 95% (light-grey), and 99% (dark-grey) of values are expected to fall—there are several values outside of the funnel, and the heterogeneity is significant. Numbers next to values refer to species identity (1, *Corvus corone*; 2, *Melanerpes formicivorus*; 3, *Philetairus socius*; 4, *Malurus cyaneus*; 5, *Vanellus chilensis*; 6, *Neolamprogus pulcher*; 7, *Prunella modularis*; 8, *Malurus elegans*; 9, *Cyanopica cooki*; 10, *Dacelo novaguinea*).

overall effect size for these is close to zero (effect size $= 0.062$, $p = 0.33$, upper and lower confidence limits $= 0.187$ and $-0.064$, $n = 7$ effect sizes from 3 species) (Fig. 2B), with very low heterogeneity ($Q = 8.85$ on 6 degrees of freedom, $p = 0.18$).

The effects of helper presence on egg size do not appear to systematically differ between instances in which helper efforts increase linearly with group size and those in which helping efforts plateau (estimate ('helping effort increases linearly with group size' versus 'helping effort plateaus') $= -0.057 \pm 0.057$, $p = 0.31$, upper and lower confidence limits $= 0.054$ and $-0.168$; $n = 19$ effect sizes from 11 studies—we excluded one study in which the relationship between helper number and helper effort was unknown). The same analysis controlling for species similarly shows no influence of how helping changes with group size on the relationship between egg size and helper numbers (estimate (helping increase linear vs plateau) $= -0.052 \pm 0.057$, $p = 0.37$, upper and lower confidence limits $= 0.060$ and $-0.164$ $n = 19$ effect sizes from 9 species).

## DISCUSSION

Our results suggest that in most studies which have examined how investment by breeding females varies with group size, females produce smaller eggs if more helpers are present. We also find that the association between smaller eggs and more helpers is particularly pronounced when females show reduced post-natal care in the presence of helpers, while egg sizes do not appear to vary with helper number when females do not adjust, or even increase, their level of offspring care in the presence of helpers. These findings suggest that concealed helper effects might occur in cooperative breeders, whereby helper

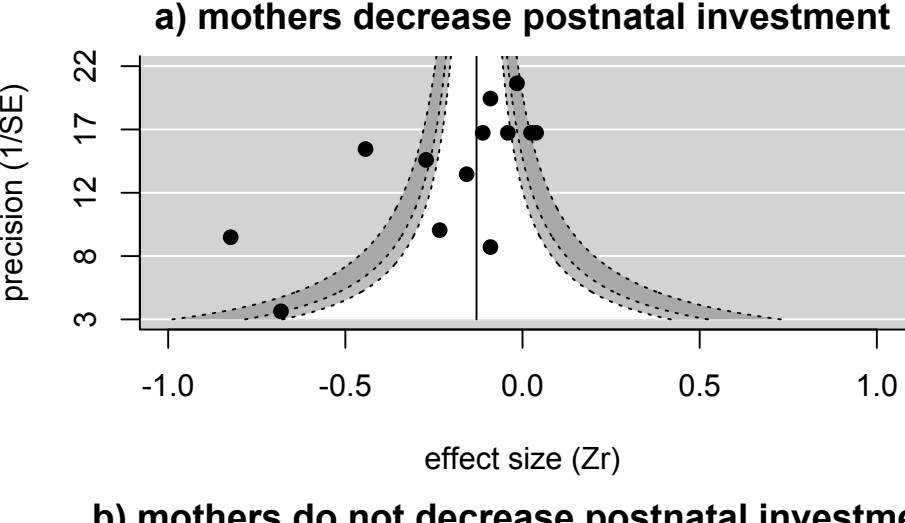

**Figure 2   Funnel plots showing the distribution of effect sizes in species where (A) mothers reduce postnatal investment in the presence of helpers (13 effect sizes from 10 studies in seven species) or (B) mothers do not reduce postnatal investment in the presence of helpers (seven effect sizes from three studies in three species).** Solid line and shaded areas indicate pooled effect size and expected values as explained in Fig. 1. For instances where mothers reduce postnatal investment (A), the pooled estimate (controlling for species) is significantly smaller than zero, with significant remaining heterogeneity. For instances where mothers do not change or even increase their postnatal investment in the presence of helpers (B), the pooled estimate (controlling for species) is not different from zero, with no significant heterogeneity among effect sizes.

efforts compensate for reduced investment by breeding females and thus provide them opportunities for load-lightening. We note that, given the small sample size and associated low statistical power, the overall effect we detected is weak and that even among these few species there is large unexplained variation in the association between maternal investment and the presence of helpers. Moreover, only a minority of studies specifically focused on within-female adjustments (3 of the 20 analyses in our dataset), and hence we cannot conclusively infer whether maternal responses are an example of adaptive plasticity rather than among-female differences. Changes in egg size could be due to other factors, such as

greater intra-group competition, although in the few detailed studies such explanations have been ruled out (e.g., *Canestrari, Marcos & Baglione, 2011*). In spite of these limitations, we explain below the insights to be gained from our study for our understanding of the benefits of helpers to breeding individuals.

Our findings are based on a small sample of correlative studies, most of which compared conditions across, rather than within, individual females. There are multiple, potentially non-exclusive explanations for why, in most species, egg sizes are smaller in nests that are tended for by a large number of alloparental carers. For example, it has previously been suggested that competition among females (*Russell & Lummaa, 2009*) or the relative costs of pre- versus post-natal care (*Savage, Russell & Johnstone, 2015*) might influence patterns of maternal investment. However, our results provide little support for the differential allocation hypothesis, as females do not appear to increase investment in the presence of helpers. Instead, female investment—both pre- and postnatal—appears to be reduced in groups with more helpers, which supports the prediction of the load-lightening hypothesis. The prevalence of effects consistent with load-lightening in our dataset could explain the common observation that helper effects on offspring size or fledging success are often weak or absent, as care efforts by helpers might be directly counterbalanced by a decrease in maternal investment (i.e., load-lightening), hence leading to little difference in offspring traits in groups with or without helpers (*Russell et al., 2007*; *Cockburn et al., 2008*; *Koenig, Walters & Haydock, 2009*; *Paquet et al., 2015*). The evidence existing thus far indicates that adaptive plasticity in maternal investment in response to helper number is at least possible. Further detailed studies focusing on variation within individuals offer the opportunity to assess the role of pre-natal maternal investment in understanding selection on cooperative systems (*Taborsky, Skubic & Bruintjes, 2007*; *Russell & Lummaa, 2009*). The scope of comparative studies could be increased by not only focusing on quantitative changes in egg size, but by also including other evidence of helper effects, such as indications of load-lightening in eusocial insects including bees (*Shpigler et al., 2013*), ants (*Villet, 1990*), and termites (*Matsuura & Kobayashi, 2010*).

We observed large variation across species in the effect of helper numbers on egg size. Differences among species might reflect that in some instances changes in female investment reflect non-adaptive processes, but even in cases where plasticity in egg size results from adaptive processes, the direction and magnitude of changes in maternal investment likely reflect trade-offs (*Savage, Russell & Johnstone, 2015*). One factor which may influence adjustment of maternal investment with a greater number of helpers (i.e., favourable conditions) is reproductive lifespan, with increased investment in favourable conditions in species with short lifespan and decreased investment in favourable conditions in long-lived species (*Paquet et al., 2015*). The occurrence of LL or DA may also depend on environmental conditions (*Hatchwell, 1999*; *Langmore et al., 2016*)—in poor conditions, females should increase investment when helpers are present in order to gain some reproductive success, because the marginal impact of care has a relatively large influence on offspring fitness. In good conditions, the impact of small amounts of additional care is low, and hence mothers would be selected to reduce care in the current generation (*Hatchwell, 1999*). Furthermore, when helpers provide assistance which increases the survival of offspring or reproductive

success beyond that which would be expected from provisioning alone (for example by reducing predation risk), breeders may be selected to increase investment in the presence of helpers (*Carranza et al., 2008*), due to the high reproductive value of the current brood in comparison with potential future broods (*Valencia et al., 2006*; *Carranza et al., 2008*; *Russell et al., 2010*; *Valencia et al., 2016*). These predictions primarily apply when maternal pre-natal investment is measured as offspring size. Different relationships might occur if changes in maternal investment are associated with offspring number: more help is likely to be beneficial when mothers increase their investment by producing a higher number of small offspring, rather than the same number of larger offspring (e.g., *Liebl et al., 2016*).

Our findings support the idea that adaptive plasticity in reproduction, with females reducing investment in current broods to prioritise future survival and reproduction, might occur in response to changes in the social environment. Further studies are needed to test whether effects hold consistently on a within-female level and whether increases in fecundity and survival are seen when load-lightening occurs. Identifying the conditions favouring the evolution of plasticity in reproductive investment has wider implications, as in many animal species the social environment is more variable than the physical environment and, indeed, social factors often mediate physical environmental factors (e.g., through foraging competition).

## ACKNOWLEDGEMENTS

Many thanks to all the authors who responded to requests for raw data: Walt Koenig, Naomi Langmore, Sarah Legge, Andrew Russell, Eduardo Santos, and Juliana Valencia. We thank Xavier Harrison, Andrew Russell and one anonymous reviewer for constructive comments on earlier versions of this manuscript.

### Funding

During the completion of this work, Sinead English was supported by a Royal Society Dorothy Hodgkin Fellowship, and Dieter Lukas was supported by a European Research Commission grant (no. 294494-THCB2011 to Tim Clutton-Brock). The funders had no role in study design, data collection and analysis, decision to publish, or preparation of the manuscript.

### Grant Disclosures

The following grant information was disclosed by the authors:
Royal Society Dorothy Hodgkin Fellowship.
European Research Commission: 294494-THCB2011.

### Competing Interests

The authors declare there are no competing interests.

## Author Contributions

- Tanmay Dixit and Dieter Lukas conceived and designed the experiments, performed the experiments, analyzed the data, contributed reagents/materials/analysis tools, wrote the paper, prepared figures and/or tables, reviewed drafts of the paper.
- Sinead English conceived and designed the experiments, performed the experiments, contributed reagents/materials/analysis tools, wrote the paper, prepared figures and/or tables, reviewed drafts of the paper.

## Data Availability

The raw data and code has been supplied as Supplemental Files.

## Supplemental Information

Supplemental information for this article can be found online at http://dx.doi.org/10.7717/peerj.4028#supplemental-information.

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
