# Peer review of "The relationship between egg size and helper number in cooperative breeders: a meta-analysis across species"

_PeerJ, doi:10.7717/peerj.4028_

## Round 0.1 · original submission · Minor Revisions

· Academic Editor

Minor Revisions

I am in agreement with both reviewers that this is an interesting study that has the potential to be a useful contribution to the field. Both reviewers have provided extremely detailed and thoughtful reviews, and have raised some comments that require attention.

In particular, reviewer 1 has noted several areas where interpretation of the results requires refinement, and suggested some changes to the analysis given the amount of data available. Both reviewers have noted areas of the discussion that require adjustment, and I would like to see these comments dealt with. The remainder of the comments pertain to opportunities to improve clarity and/or the accuracy of referencing, and so should be straightforward.

Please also move Fig. 1 to Online material as suggested by reviewer 2, as I think your methodology describing study selection is perfectly clear and doesn't require this figure in the main text.

I look forward to seeing a revised version.

Reviewer 1 ·

Basic reporting

Strong. See below for some points to consider.

Experimental design

Strong. See below for some points to consider.

Validity of the findings

Revisions are needed to bring the conclusions in line with the results.

Comments for the author

This is an interesting paper that presents a novel meta-analysis of whether egg size correlates with helper number in cooperative breeders. This relationship would be predicted if, for example, mothers plastically adjusted their egg size according to helper numbers. This is an issue of broad current interest in the cooperative breeding and maternal investment literature. While the study struggles with sample size limitations it does provide a novel contribution to knowledge in this area. Revisions could be usefully made, however, to better align the overall conclusions with the findings, increase transparency regarding the use of repeated measures of species, clarify a few points, and provide some more inclusive coverage of the relevant literature. I cover these points below.

KEY POINTS

1. Greater caution is needed with interpretation, as the meta-analytical findings do not currently provide direct evidence of maternal plasticity or helper effects.

Evidence that egg size varies with helper number does not necessarily reveal (i) that mothers actually show plasticity in egg size (as the observed variation in egg size could arise from among- rather than within-female variation in egg size, and while some studies considered this many did not), or (ii) that this variation, whether plastic or not, is causally attributable to variation in helper number (as there could well be confounds of helper number; e.g. territory quality).

The manuscript would therefore benefit from…
- Acknowledgement of both points in the discussion
- Attention to the potential alternative explanations for the findings that recognition of these points highlights
- Adjustments to the phrasing of arguments and conclusions throughout the manuscript regarding plasticity and effects of helper number (including the abstract and title, see point 2).


2. The title needs to be revised to more acurately reflect the findings.

The title “Cooperatively breeding mothers reduce their investment in eggs in larger groups: a meta-analysis across species” currently seems misleading for two reasons:
(i) the statistical evidence for such a correlation across all species falls shy of the significance threshold once repeated measures of species are controlled
(ii) the meta-analysis is not based exclusively on within-female relationships between helper number and egg size (and so the meta-analytical result could be a product principally of among-female variation in egg size rather than within-female plasticity; see point 1).

A more general title based on the question rather than the answer would resolve this easily. E.g. “Investigating egg size variation with helper number in cooperative breeders: a meta-analysis across species”

3. Please clarify the classification of species.
In Supp Table 1, it is not currently clear why, when using the data for Malurus elegans from Lejeune et al 2016, the species has been classified as showing compensatory care for the within-mother effect sizes but additive care for the among-mother effect sizes. Surely the species either shows compensatory or additive care? Indeed, why use among-female effect sizes at all where within-female effect sizes are available, if the goal is to study plasticity? Please clarify / resolve.

4. Please adjust the figures to increase transparency regarding the repeated measures of species.
In the funnel plots for Figures 2 and 3 it would be great if the authors could place a number next to each data point indicating which species it comes from (there aren’t many so this should work well), and then have the numbered list of species in the figure legend. This will allow the reader to consider the extent to which any apparent pattern is influenced by the inclusion of repeated measures of species.


SPECIFIC POINTS

INTRODUCTION

5. It would be good to see the introduction better address two relevant areas of the literature:
(i) The introduction currently skates over the rather intensive research effort to date on load-lightening of post-natal maternal investment in cooperative breeders. Given the parallels with the questions asked here (re the impact on pre-natal maternal investment) it would be good to more fully explain the advances in this area to date (in particular the relevant insights from the comparative work in Hatchwell 1999 – currently only cited in passing).
(ii) It also seems odd to cite virtually none of the empirical literature addressing maternal plasticity in egg size in cooperative breeders, when this is the focus of the manuscript.

6. Lines 78-81 – please clarify the argument here

7. Lines 102-105 - It seems odd to justify the study’s focus on pre-natal rather than post-natal maternal investment on this basis when …
- Its not clear why maternal investment in egg size “cannot be directly influenced by helpers”, as helpers do have the potential to impact maternal condition at laying (e.g. via effects of group size on vigilance/foraging trade-offs etc.), and
- Presumably the main reason for focussing a meta-analysis on pre-natal investment in the egg is novelty (much work has already been done on post-natal maternal load-lightening)?

8. Lines 105-109 and in the paper’s Summary
Please clarify the definition of ‘additive’ and ‘compensatory’ care. As set up by Hatchwell 1999, doesn’t ‘additive care’ refer to a scenario where the mother does not compensate for the contributions of helpers (and ‘compensatory care’ refers to a scenario where she does)? The wording in this manuscript suggests that the distinction between additive and compensatory care here is made on the basis of whether the helpers (rather than the mother) compensate for each others’ contributions. This is subtly different, and presumably has implications for both the logic and the classification of species. Please clarify.

9. Lines 196-197 – please remove ‘broad taxonomic spread of species (fish to birds) in our sample’ from this argument or rephrase, as this doesn't really help to justify not controlling for phylogeny (there is only 1 fish).

METHODS

10. Lines 119-122 – The definition of ‘cooperative breeder’ needs a clearer definition of ‘helper’. The inclusion of dunnocks (presumably polyandrous?) in the analysis, for example, suggests that the authors are comfortable with including species in which the helpers are actually breeders. What then is a ‘helper’? To increase transparency and clarify the logic it would also be good to explicitly acknowledge here and (briefly) justify the inclusion of cooperatively polyandrous species.

11. Conducting a statistical test for a difference between species with additive and compensatory care strategies seems to me to be pushing the data too far - there are data for only 3 species with compensatory care (perhaps 2 depending on the answer to point 3 above), so such a comparison at present seems pretty meaningless and potentially misleading. Wouldn’t it be more appropriate to simply plot the current figures and discuss the patterns by inspection? It would still be good to follow this up with an analysis of the overall data set restricted to species with additive care only (as there are at least 8+ species of this kind), which yields an interesting result, but then also conducting this analysis on the (just 2 or 3) compensatory species alone also seems to be pushing the analysis too far.

RESULTS

12. To enhance transparency regarding the data structure throughout, whenever quoting sample sizes it would be good to break down the sample size in to the number of effect sizes and the number of species. So rather than reporting “n = 20” report “n = 20 effect sizes from 10 species”.

DISCUSSION

13. Line 290 – rephrase – this isn’t ‘demonstrated’.

14. lines 291-293 – it seems a little late to acknowledge this critical point. To maximise transparency it would be good to include this point in the 1st paragraph of the discussion where the findings are summarised.

15. The final paragraph of the discussion goes a little hyperbolic – please bring the claims in line with the findings (greater caution needed throughout).
e.g. Rephrase “The prevalence of LL indicates adaptive plasticity”. A prevalence of LL would indicate plasticity but such a prevalence has not been demonstrated here (as the analyses haven’t focussed on within-female relationships / experimental work) so a more cautious phrasing is needed.
e.g. “our findings demonstrate the pervasive influence of the social environment on life-history” – that would be nice, but they don’t - the effects, if they do reflect plastic responses, could presumably be driven instead by correlates of the social environment (see point 1).
e.g. “Our results demonstrate the importance of plasticity in maternal reproductive investment in cooperative breeders” – same issue.

·

Basic reporting

This is an interesting study on an important topic that is gaining increased interest. I have a lot of comments, mainly editorial, and so easy to deal with, but important nonetheless. The key issues are: (1) Many references from studies of non-cooperative breeders are used in inappropriate places to support points that are particular to cooperative breeders; (2) In many cases were references to cooperative breeding studies are made, they pertain to provisioning and not egg investment, when in this case the point is not necessarily transferable from nestling to egg investment; and (3) As a consequence of each, key information and or references are missing on the theory of egg investment in cooperative breeders (see Savage et al. 2013a, b and 2015) (these references are provided, but key theory presented within is generally missing (e.g. Carranza et al. 2008; Russell & Lummaa 2009). I have provided as much help as I can to address these issues. You might not agree with all, and that is fine of course, but the overarching point is that more theory on egg investment in cooperative breeders needs to be introduced to put your study in context and acknowledge the growing background in this area, and I personally think you need to be quite careful with many of the references you use as they are not necessary transferrable to the point being made, and so are either confusing or mis-leading.


Line 65: Other possible references include Metcalfe and Monaghan 2001 and Lummaa and Clutton-Brock 2002, both in TREE. I mention these because they really deal with either other aspects of fitness (the former) or long-term, intergenerational effects (the latter). As such, I feel that both add to your reference list, while at the moment, I feel that the references are really saying the same thing.

Line 66-67: I think you are making an empirical point, but mix this up with references of a theoretical nature (e.g. Maynard Smith 1977). To me, what is really needed here is some clear experimental evidence of a cost of not just reproduction, but increased effort.

Line 71: Few is subjective, so I would suggest you use Fewer instead.

Line 73-74: The word excellent is subjective (maybe switch to apt or appropriate), and the subject of the sentence needs to be made clear. At the moment, I think you are trying to carry the subject from the previous paragraph, but it is not so clear.

Line 75: Delete in a single location as this is not true. Meerkat pups following the groups are not in a single location, and nor are the offspring in the numerous plural cooperative species in which helpers provide care to offspring from multiple nests.

Line 78: The Hatchwell 1999 reference is fine for this, as it is a review, but the komdeur 1994 reference is a bit random. First it is subsumed within Hatchwell. Second, it is not the first evidence. Third, it is not the best evidence. Either you want to provide the first suggestion and/or the best, latest experimental evidence (preferably both) followed by a review, but not a random paper.

Lines 78-81: Several lines are missing here, as you go from something super-general to suddenly in the next sentence talking about eggs in birds. In line 78, all we know is that you are talking about cooperative breeders, maybe in birds, and something to do with investment. But in line 79 you are on eggs in birds. You need several intervening sentences. First, you need to say that theory suggests that mothers should change investment according to workforce (e.g. Crick 1992; Carranza et al. 2008; Johnstone 2011; Savage et al. 2013, 2015; McAuliffe et al. 2015). Second, you need to say that experimental tests have been largely conducted during provisioning (e.g. Hatchwell & Russell 1996; Wright et al. 2008; Russell et al. 2008). Finally, you need to raise the question about whether or not such effects should be also manifest at the egg stage (Savage et al. 2015 deals with this directly).

Line 84: Not sure you need to call this load-lightening or maternal load-lightening, this is kind of the same thing. Koenig et al. 2008 termed the specific hypothesis at hand as the concealed helper hypothesis.

Line 85 and Lines 88-89: Not sure either Crick or Heinsohn dealt with egg investment, the more appropriate references here are Russell and Lummaa 2009 and Savage et al. 2015.

Line 87: Lessells, has 2 l’s

Line 91: Again, do any of the references deal with egg investment, again the more appropriate references as outlined above. Personally, I think there is a very important distinction between reducing investment at egg versus chick stages, and the references used throughout should acknowledge this difference.

Line 94: You need to be careful here. The Burley hypothesis is distinct from the Sheldon one, and both are distinct from the Russell and Lummaa one. The Burley reference is actually more akin to load-lightening (in a weird way because she allows her mate to load-lighten when high quality), than to Sheldon’s definition of differential allocation, and Russell and Lummaa 2009 expand on the Sheldon definition to make it appropriate to cooperative breeders.

Line 96-97: Maybe not here, but you should acknowledge in the discussion that it is conceivable that reduced egg investment with increasing group size is non-adaptive, and results from increased foraging competition. There are some nice studies that can rule this out, e.g. Canestrari et al. 2011.

Line 101: Personally, I would set this up as a direct test between the concealed helper hypothesis/load-lightening and differential allocation, rather than a simple description of how mothers react to increasing helper numbers. Russell and Lummaa 2009 as well as Savage et al. 2015 clearly set out some predictions. The former suggest that DA might be predicted in plural cooperative breeders, or other cooperative breeding social structures that generates competition among reproductive females within groups. Savage et al. suggested that DA versus LL is expected as a function of the costs off egg laying, with LL occurring when the costs of egg production are high, and DA when it is low. In other papers, Savage et al. 2013 (BES) also suggested that it will vary as a function of helper relatedness. Further, Carranza et al. 2008 also set out circumstances in which DA can arise. In saying all this, I am not suggesting that you can provide a full test of these hypotheses, but it is important to at least provide the background theory, and the point of your study. I think it would be fine to say that the primary is to shed light on these hypotheses by testing whether helpers generally cause LL or DA effects.

Line 103: I would not say that egg size cannot be affected by helper number. You can have evolved responses, and in meerkats, helpers influence maternal condition, through allowing her to reduce investment in pup feeding, which in turn positively influences investment in neonates (see Russell et al. 2003 BE). In some bird species, helpers feed females before egg-laying, which might also impact their condition. I think would be okay to say that they presumably have reduced direct effects in egg compared with offspring, but whether or not direct effects are not yet known.

Lines 107-109: I don’t understand this, nor the reference to Langmore et al. 2016.

Lines 118-119: This is not strictly true, usually only the female incubates the eggs, I suppose you mean provisioning, but need to be more specific.

Lines 122-126: There are plenty of non-eusocial insects where the above constraints are lifted. The main problem with insects for your purposes is that there are almost no studies of egg size across helper numbers. Russell & Lummaa 2009 could not find any non-eusocial cooperatively breeding insects in which maternal investment in egg size as a function of helper number was assessed, but it would be worthwhile checking to see if there are any since this publication, which would help bolster your sample size.

Lines 154-158: Sorry about this, I was in the middle of a high load of teaching. I can confirm to you and to the other reviewers and editor, that categorisation of helper effects according to the presence versus absence of helpers is valid in wrens as the number of groups with greater than one helper were very limited. Nevertheless, categorisation will inevitable inflate your effect size estimate. I will provide you with the necessary details, although there will be no qualitative change to your results.

I am not sure Figure 1 is necessary, perhaps it could go online?

Line 268: I think it would be nice here to re-iterate that this results provides general support for the concealed helper hypothesis.

Line 271: It might be nice here to re-iterate that this evidence complements the hypothesis that additive care by helpers should lead to LL in females in terms of provisioning (Hatchwell 1999).

Line 280: You need to be a wee bit careful here, as Russell et al. 2007 found the evidence and came up with the idea, but Koenig et al. which did not find any supporting evidence, coined the hypothesis. At the moment, you use the two citations to different ends, but this is not clear, making it suggest that the Koenig et al study is also supportive.

Line 286: At the moment, you use the term In contrast, to contrast with LL, but the subject of the previous sentence is about support not LL per se. It therefore sounds, as written, that you are going to provide contrary evidence for LL, but in fact go onto provide contrary evidence for DA. I think I would suggest starting a new paragraph, and making the subject of the sentence clear.

I also think it would be appropriate to discuss the suggestion by Russell and Lummaa that DA should be more likely when reproductive females compete within groups, since their fitness will be impacted by the relative success of their offspring. It is at least noteworthy, that the majority of the species that you studied were singular breeders, meaning females do not compete reproductively amongst each other within groups.

Line 311: I think that this Savage et al. 2015 reference should be cited and discussed in line with your results, but this is the wrong citation here, and should be Carranza et al. 2008.

Experimental design

No comments

Validity of the findings

No comment

Comments for the author

I have added everything above under basic comments

---

## Round 0.2 · Major Revisions

· Academic Editor

Major Revisions

Your manuscript has now been re-assessed by one of the original reviewers, and you will see that they have raised several issues with the current version.

In particular, I agree with the reviewer that the issue of causality in the relationship between post-natal helper and pre-natal maternal investment is unresolved, and so the rationale for the analyses in the current manuscript as presented is uncertain.

The reviewer has provided a very detailed set of comments, and I would like to see all comments addressed in the revision.

·

Basic reporting

see below

Experimental design

see below

Validity of the findings

see below

Comments for the author

I am very sorry to say that i am disappointed by this latest version of the MS, and again have spent at least a full day refereeing it.

In particular, I still have quite a few significant issues with clarity and citation use, which to me are very linked in this MS. I said this last time, but perhaps I was not clear. For example, there are many examples of using the wrong citations or using a mixture of citations that confuse rather than clarify. When this occurs, it is unclear whether the point being made is wrong and the citation correct, or vice versa. Either way, citations should help clarify a point, not confuse it.

A key issue is the use of citations based on theory of post-natal care to support a point made about pre-natal investment (e.g Crick 1992, Hatchwell 1999 or even Brown 1987 or Stacey & Koenig 1990). There are many reasons why one should not do this. Most notably, mothers might reduce investment in offspring provisioning because chicks are less hungry due to the contributions of other group members and consequential effects on offspring begging. By contrast, given that mother must by definition lay the eggs before the contributions of any others, there are obvious reasons why load-lightening at this stage might not be expected, despite showing post-natal LL. This is why LL at the egg stage was not thought of prior to 2007. It is incorrect, presumptuous and mis-leading to use Crick 1992 or Hatchwell 1999, to support the load-lightening hypothesis at the egg stage.

Using theory provided to understand post-natal LL as a means to understanding pre-natal maternal strategies (e.g. by citing Hatchwell 1999) leads one inevitably to conclude that it is the post-natal responses to variation in helper number that drive pre-natal maternal investment strategies. By contrast, theory on maternal investment pre-natally make no such assumptions (see Russell et al. 2007, Taborsky et al. 2007, Russell & Lummaa 2009, Savage et al. 2013, 2015). Instead, in the vast majority of the hypotheses or theory provided in the above papers, the point concerning LL are more simple: an increase in helper numbers leads to an expectation of more food (or protection) for offspring, which allows mothers to save resources at the egg stage without losing fitness. The authors are right that if all co-carers fully compensated for each other’s presence, this assumption falls away. But there is next to no evidence that this is the case. Instead, any compensation is partial, and so increases in helper number (or helper presence versus absence) generate additive care. Further, more detailed theoretical attention, particularly by Savage et al., shows that post-natal care responses, are themselves linked with pre-maternal investment. This means that it is much safer to use helper presence/number as a predictor of maternal pre-natal investment (as is typical) rather than helper behaviour (this study). Thus, I am happy with the first analysis on helper effects on maternal investment, but less happy with the suggestion that helper behaviour post-natally drives this pre-natal investment (since it could be the other way around).

My worry about the second analysis is exacerbated by the lack of clear definition of compensatory versus additive care. For example, dunnocks appear to be used as an example of compensatory care (Discussion), but increasing group sizes are associated with additive care in this species, because compensations are incomplete. I would like to see a table inset in this MS, which outlines the species used as compensatory or additive with a clear justification for each one. I think that what we will find is that there is a continuum, from rarely fully compensatory to highly, or even rarely, super-additive care. Finally, on this point, I worry that because of the rarity of fully compensatory systems (perhaps absence) the question of whether additive care post-natally is collinear with the first question?


Below I outline my numerous specific comments

Line 72-73. As currently written this statement is untrue. Not only are there ca. 100 studies in cooperative breeders looking at parental responses to variation in the social environment, but variation in the social environment includes variation in co-carer contributions and so includes bi-parental care systems. There are over 100 more studies in this care system. I am not sure this is fewer than the number of studies looking at variation in response to other traits, and even if it is, the point is moot.

Lines 74-75. As written, this includes bi-parental care. I think it is fine to include bi-parental care at this point, but to get to cooperative breeders you need to say how they differ from bi-parental care systems, and the answer here is that they have variation in the number of carers. This allows maternal variation in investment to be linked to variation in the social environment, without direct manipulation of that environment (ie in cooperative breeders the social environment varies among mothers, which is not so obviously the case in bi-parental care systems).

Lines 75-79. The point that needs to come across here is not so much what cooperative breeders are, but that there is substantial variation in helper number within and among species, allowing tests of maternal responses. This is one of the key points made by Russell and Lummaa 2009, which formed the basis for their predictions about how this variation might be expected to affect maternal investment at the egg stage.

Lines 80-84. You need to be careful here. As the other Reviewer pointed out last time, you cannot use words like plasticity, unless you show within-female variation, and the only way to show this is to do an experiment or to use long-term data which allows sufficient contrasts. This is a classic case of where your message might be correct, but most of your citations are inappropriate, or whether you citations are appropriate but that the point being made need not be restricted to plasticity. Given the sentence before, I am assuming that the statement is correct, but most of the citations are not, including Hatchwell et al. 2004, Heinsohn 2004, Koenig et al. 2011, as none of them use experiments to specifically detail within-parent responses. Cockburn et al. 2008 provides comparisons of long-term data to show adaptive female, but not male responses in superb fairy wrens. One of the Macdonald et al. bell minor papers also uses nice removal and chick begging experiments to test for plasticity in bell minors, and one of these should also be cited.

Lines 84-89. This is a good example where multiple points are made, but all the references are added to the end – making it impossible to know which substantiates which point. To make matters worse here, the points are incorrect, and all of the references are inappropriate. First, you are talking about responses here, and so by definition the consequences of experimental manipulation or very carefully detailed statistical analyses based on longitudinal data. There are only a handful of such studies, listed above (none cited in this section), but I am not sure they are sufficiently numerous to talk about a typical response. So, instead. I would suggest you say…One such response is…..

Second, if you are somehow trying to talk more generally, you cannot use the word response, it would not follow your previous sentence and again you cannot really talk about a typical response (see Hatchwell 1999). In this case, the Reyer and Manica references are irrelevant, but at least Hatchwell and Kingma are good. Either way, you need to be more careful in your use of words, and more precise in both what you say and how you cite the work.
Line 82: Manica and Mirani reference is a poor choice, there are many confounding factors in that paper and compensation cannot be inferred. Compensation can only be shown by experiment and these have been done rarely, e.g. Hatchwell & Russell 1996, Wright 1998, Macdonald et al. in a series of bell minor papers; Russell et al. 2008

Line 92: Neither Brown 1987 nor Koenig and Stacey have anything to do with eggs and should be deleted. Canestrari et al. should also be removed unless you want to cite every single paper that is in your meta-analysis. The only paper that outlined the idea and made any theoretical predictions is Tabosrky et al. 2007 (for protection) or Russell et al. 2007 (provisioning) and Russell & Lummaa 2009 (inclusive of both and other helper effects); and then the formal models of Savage et al.

Lies 89 & 96: You have Savage et al. 2015, then Savage, Russell and Johnstone 2015, these formats need to be reversed (full authors first and then et al. 2nd)?

Line 101. Crick 1992 makes no mention of eggs. I said this in my last review: one cannot extend theory on responses to offspring provisioning with responses on eggs. They are many reasons why load-lightening at the egg stage is very different, including the fact that no other group member in real time can respond. The only theoretical treatment of helper effects on egg investment are Taborsky et al. 2007, Russell et al. 2007, Russell & Lummaa 2009, and the series of Savage et al. papers; particularly his 2013 and 2015 papers. No other paper is relevant that I know, and certainly not those cited.

Line 103-104: Remove Canestrari et al. since this is entirely empirical and does not extend any of the points made previously. And Santos 2016 is a thesis, which should be removed completely.

Line 111. Savage et al. 2015 should probably cited here which squarely considers how mothers should invest with increasing helper numbers.

Line 116. Remove the word “exists”

Lines 117-119. Not sure what is trying to be said here. If I understand correctly, does a meta-analyses really indicate whether detailed studies are required to understand selection and the biology of individual species?

Lines 119-122. I am not sure of your logic here. I am not sure you need to justify why you look at pre-hatching rather than post-hatching investment. They are totally different questions. I think the important point is more because it is a stull relatively under-studied area of research, with demonstrable implications for understanding selection on cooperative systems (e.g. Russell and Taborsky 2007 refs).

Lines 125-128. Sorry, I have no idea of the logic here. I suggest that the authors check out Russell & Lummaa 2009 and Savage et al. 2015, for logic in when we expect LL of DH at the egg stage. Again the authors cite a paper that has nothing to do with egg investment, and so confuse the point being made. The problem with understanding post-natal investment is that maternal, paternal and helper contributions are dynamic and potentially negotiated as chicks age (McNamara et al. 1999). Post-natally - one cannot say categorically that mothers are more likely to LL when helper effects are additive, because the reverse might be true, helper effects might be additive because mothers load-lighten. The interesting thing about the egg investment stage, is that mothers need to invest before partners or helpers are able to respond. Indeed, the response of co-carers, when it occurs, is itself a part-function of maternal investment at the egg stage (Savage et al. papers). Thus, it is circular, mis-leading and potentially wrong, to use co-carer investment strategies at the chick stage to predict maternal investment strategies at the egg stage. Predictions about LL versus DA at the egg stage need to be articulated from papers that deal specifically with these points at the egg stage.

Line 130. Need to say at the egg stage

Line 131. Following my comments above, you need to be careful to define what you mean by additive care. You mean at the chick stage, and you mean of all co-carers other than the mother? Or do you mean overall, irrespective of LL by the mother at the chick stage. You state in the methods that you simply used what the authors had reported, but 1-2 lines summary of this is required in the Intro and a table in the methods. For example, in chestnut-crowned babblers, mothers reduce their provisioning contribution to almost zero in large groups. But, because the contributions of male carers are unaffected by the number of co-carers, and because the number of co-carers can be numerous, there is a positive association between carer number and overall provisioning rate of the brood. I am guessing that you would call this additive care, and I would too, but serves to illustrate the point that not all individuals in the group need to show additive care for care to be additive. Indeed, you could also have a situation where all group members LL, but care is still additive, because LL is only partial. Finally, it is unclear whether you are saying that because care is additive, we expect maternal LL at the egg stage (which is what I think you are saying), but I would say that opposite is also possible, additive care at the nestling stage arises because of maternal LL at the egg stage. You do not need to make all these points of course, but I hope you can see that some clarity is required, and you need to be clear that any detected association between LL at the egg stage and additive care at the nestling stage could arises as a cause or effect.

Lines 142-146. I fundamentally disagree with this comment, and have predicted elsewhere (Russell & Lummaa 2009) that maternal effects in social insects should be much stronger than in vertebrates. All you need to say here is that the data is lacking.

Lines 146-149. This is also incorrect. Broods often receive more incubation and more food in communal birds because the number of nest attendants is far great than the usual 2. The problem here is that you cannot identify which egg came from which mother, and so it is difficult to know what causes any changes in egg size in communal nesters.

Line 209, were they not also inclusive of different data, which I would have thought would have been more important. For example, Langmore et al., published 9 years later, used far more years of data than Russell et al..

Line 273. Helper effects can be additive even if they are compensatory. Please can you clarify if these species show additive care, albeit perhaps to a lower level than others species.

Lines 288-289. Need to specify that less means reduced egg size not clutch size

Lines 293-294. Is this association not just an epiphenomenon of the fact that there is an association between helper number and maternal egg investment across species, and almost all of the species included showed additive care?

Lines 294-298. I don’t understand what is being said here. Dunnocks show additive care, females with 2 males receive more food for their young than those with only 1 male. This additive are arises from incomplete compensation, and females increase clutch size as a consequence (Davies and Hatchwell 1992). As a consequence, I do not follow the logic of your point – females increase egg investment and decrease provisioning when breeding polyandroulsy; so is the decrease in provisioning because of the extra male or because of their increased egg investment, or a bit of both?

Lines 311-315. Please at least acknowledge that causality is unknown. We do not know whether maternal LL at the egg stage generates additive effects at the chick stage, or not. I do not necessarily disagree, but we do not know the direction of causality at this time.

Line 317. Did Woxwold and Magrath not suggest that the relationship between group size and breeding success might be weak because not all group member contribute? If so, this is a very different argument to the one you present here, which was really the foundation of Russell et al. 2007, and then furthered by Koenig et al. when he coined the term concealed helper effects.

Lines 340-345. I do not disagree with any of this, but it is at least noteworthy that this analysis does not control for maternal egg investment, and so any role of maternal egg investment (particularly clutch size) in generating differential nestling starvation (Liebl et al. 2016 Anim Behav).

---

## Round 0.3 · accepted · Accept

· Academic Editor

Accept

Both the reviewer and I agree that this recent version of the manuscript is greatly improved after your substantial revisions, particularly with regard to the streamlining of analyses. I look forward to seeing this article in print.

·

Basic reporting

Great

Experimental design

Great

Validity of the findings

Great

Comments for the author

Very well done for dealing with all of my previous comments, I do think the MS is now far clearer and more informative than before.